# A New Approach to Dynamic Anthropometry for the Ergonomic Design of a Fashionable Personalised Garment

Manuela Lacramioara Avadanei [1,*,†], Sabina Olaru [2,3,*], Ionut Dulgheriu [1], Savin Dorin Ionesi [1], Emil Constantin Loghin [1] and Irina Ionescu [1]

1   Knitting and Clothing Department, Faculty of Industrial Design and Business Management, Technical University "Gheorghe Asachi" of Iasi, 29, Prof. Dr. Doc. Dimitrie Mangeron, 700050 Iasi, Romania; ionut.dulgheriu@yahoo.com (I.D.); dionesi@tex.tuiasi.ro (S.D.I.); loghin.emil@gmail.com (E.C.L.); iirina@tex.tuiasi.ro (I.I.)
2   National Research Development Institute for Textiles and Leather, 030508 Bucharest, Romania
3   The Executive Unit for the Financing of Higher Education, Research, Development and Innovation (UEFISCDI), 010987 Bucharest, Romania
*   Correspondence: manuela.avadanei@gmail.com or mavad@tex.tuiasi.ro (M.L.A.); sabina.olaru@incdtp.ro (S.O.)
†   Member of the ErgoWork Society in Romania.

**Abstract:** Background: A challenge for designers is to create fashionable and very well-fitting personalised garments (multi-layered) that have a suitable shape (balance and size) and provide the wearer with the desired degree of freedom. In this paper, the authors have developed an ergonomic solution for designing the pattern of a business casual men's jacket by integrating the dynamic data of the body into the design process. Methodology: The pattern was elaborated by interactive design process based on mathematical relationships and the use of specific input data. The 3D virtual prototype was created in Clo3D (the static and typical dynamic positions of the mannequin). The dynamic data needed for the study were measured directly on 50 male subjects. These values were analysed by using the statistical method and then integrated into the design scenario in a specific way. The shapes of the new 3D prototypes were evaluated by examining the relationships between the constructive and longitudinal allowance along the back region as independent variables and sleeve angle and upper back tension as dependent variables. Results: By allowing a certain degree of dynamic effect in the design process, one can see that the personalized model of the casual business jacket with Ab (constructive allowance) = 4.5 cm and Aars (longitudinal allowance distributed along with the back height) = 3.6 cm is well balanced and fits the body. Conclusions: This design method can be used and further developed for other garment categories and customers by any design department that has the right IT tools to facilitate the personalized design process.

**Keywords:** ergonomic design; dynamic anthropometry; virtual prototype; personalized garment

## 1. Introduction

The digital transformation of the fashion and clothing industry has brought benefits and given rise to challenges for both manufacturers and customers. Depending on the degree of complexity and destination, the new product/model is designed digitally, verified in a virtual environment (to validate its design), and then manufactured and advertised. The virtual prototype of the model can be obtained by designing the parts in the product structure (using 2D tools of CAD systems) and then importing them in a 3D environment for virtual simulation or by directly designing the components on a mannequin (3D design), and then extracting them by unfolding and interpolating the shape contour lines.

Digital tools are also widely used by customers in order to customize different products or services according to their needs and then to purchase them at an acceptable price. The demand for product customization is one of the greatest challenges facing the

consumer goods industry nowadays, including fashion. The diversity of models, styles, trends, and especially the customer's requirements give rise to a big problem for manufacturers/designers, namely that of manufacturing products that best fit the shape of the customer's body, to meet their requirements, at the lowest possible cost and most importantly, in the shortest possible amount of time.

The spatial configuration of the clothing product is a complex one and is the final result of a design process (in which information about the shape of the customer's body and the stylistic details of the model are integrated), and a careful selection of raw materials and materials is necessary for its realization and manufacturing. The spatial shape of the multilayered clothing products (products with shoulder support), highly adjusted on the body, is designed in successive stages (through virtual and physical prototypes), because the appearance of the product, its placement on the customer's body, and the geometry of the model lines are influenced by the properties of the textile materials (the layers interact with each other, they lie on the complex spatial geometries of the human body, and the cut-out parts have complex shapes). For product models in this category, the designer must develop the best solution and integrate the customer's requirements into the conceptual development of the product, while also keeping in mind that the product must provide the wearer with a certain degree of comfort (freedom of breathing and its normal dynamics).

Freedom of movement is an essential and important requirement in the case of clothing and should be addressed separately for fashion products, personal protective equipment, or sports gear. For products belonging to the last two categories, freedom of movement is ensured by design (quantifying the dynamic effects of the dimensions of the body using mathematical relations to determine the dimensions of the surface of the product elements, and by using optimal amounts of constructive additions), constructive elements (vents or folds), and by combining materials appropriate to the final purpose of the product. These products must allow the wearer to perform repetitive movements of relatively constant amplitude for a certain amount of time; if they are personal protective products, they must protect them from the action of environmental factors, and if they are sports gear products, they must allow the wearer to perform at an optimal level.

In the case of fashion products with a very tight and adjusted silhouette (especially for multilayered products), the degree of freedom of movement is conditioned by the values of the constructive additions used in the geometric design of product elements, as well as by the number, size, and shape of the constituent parts, and the manufacturing technology. One could argue that there is a conflict between the need to have a light product (for comfort in wearing) and the one for a tight product, which closely resembles the shape of the body (according to the customer's requirements). This apparent contradiction can be resolved by developing solutions for an ergonomic and customized design of the components, using digital tools specifically tailored for this purpose.

Applying ergonomic principles to fashion design is a significant challenge, as it combines the garment characteristics with human technology to form an unbiased instrument. Ref. [1] suggested that "clothing adapts to man", as opposed to the outdated concept that "man adapts to clothing". Instead, ergonomic designs recognize the harmonious relationship between the human body, clothing, and their environment as well as the importance of "beauty in the form" and "efficiency" in clothing design. It analyses each index of the clothing medium and the various requirements of the human body to achieve a satisfactory balance of comfort, wellbeing, safety, fitness, function, appearance, and personality.

Ref. [2] studied the development of new strategies for workwear design. Their study analyzed the activities of workers in the electrical sector. The results identified the constraints and the variables that contributed to the development of an analytical model for the design of ergonomic workwear.

The information in the literature [3,4] refers to the design of products used as work equipment. In this case, the ergonomics study begins in the design phase and continues in the operational phase through corrections and optimizations. Through subsequent

corrections and optimizations, the ergonomics certificate is obtained, an instrument that establishes the working conditions for the regular use of the designed product.

Ref. [5] developed an ergonomic evaluation system for the design of partial pressure suits for high altitudes (PPSs). An ergonomics index was created to evaluate the performance of different suit designs with and without pressure. The resulting ergonomics index provided a theoretical basis and practical guidance for mission planners, suit designers, and engineers involved in equipment design, and assisted in evaluating partial pressure suits.

Before the advent of ready-to-wear clothing (RTW), garments were manufactured based on the body measurements of individual users (anthropometry). This indicated that ergonomics has always been an integral part of garment manufacturing; however, the concept of ergonomics was not mentioned in the context of fashion engineering and design. Ref. [6] emphasized the need for an organized framework for ergonomics that would be suitable for education and research in the field of fashion and concluded that the following would be necessary: (1) the five aspects of ergonomics; (2) anthropometry and biomechanics; and (3) the three domains of ergonomics.

Bibliographic research [7–9] has shown that the use of ergonomics for the design of fashion products has been imprecise, and the study and application of product ergonomics has typically been inconsistent.

So far, the scientific publications in the field, including Refs. [10,11] have recommended using the comfort parameters of monolayer products to evaluate the pressure they exert on the human body. However, the matter of designing patterns for personalised ergonomic multilayered fashion products with an adjusted silhouette while taking into account the data concerning the dynamics of the human body, has not been addressed for fashionable garments.

The key questions that we had in mind in this study were whether ergonomics can be applied to fashion and if ergonomic design is important for fashion products.

The objective of this study was to develop a solution for the ergonomic design of a fitted (casual business) jacket model based on its two-dimensional, geometric pattern while integrating data on the dynamics of the human body to improve the functionality and comfort.

In this paper, the authors propose an ergonomic and personalised approach for designing the patterns for a men's jacket model with a fitted silhouette (business casual outfit) using digital tools specifically developed for the geometric design of textile clothing (the Made-to-Measure working environment of the Gemini system CAD). The customer's and model data are integrated into the geometric layer of the CAD system, and then mathematical relations are used to define the positions of the important points on the outline of the pattern, which are in turn connected by straight or curved lines and modelled according to the requirements of the product. If one alters the values of the initial data and the structure of the mathematical relations used in the design scenario, corresponding new patterns will automatically be generated.

The presented algorithm can be used in the creation departments of clothing companies that use a CAD system with specific functions for the geometric design of the components of their products. It gives designers the possibility to create new models in response to customer requests quickly, diversify/personalize the range of the ones they launch on the market, reduce the validation time of the new prototypes, and experiment with the most complex design solutions.

This design approach can be considered sustainable for the following reasons:

- The garments have an optimized surface, which is a result of ergonomic design. The length of the model markers is reduced, and the percentage of used material is maximized to the best possible extend;
- By taking the customer's body measurements, the dimensions of the model and by following ergonomic principles, the product will look well on the customer's body (balance and fit) and provide the desired comfort. In this situation, the customer's

degree of satisfaction with the purchased item is going to be high and the number of items that do not fit the customer is reduced (these items are considered waste);

- By testing and evaluating the new model in a 3D virtual environment on the personalized avatar (which reflects the customer's measurements) in static and dynamic positions, the design of the new model can be validated, and the number of physical prototypes is drastically reduced, which in turn means that the necessary amounts of raw materials, energy, and chemicals are significantly diminished;
- This method can be used for developing block networks for different garment categories. If necessary, these blocks can be easily adapted to the customer's requirements and the characteristics of the models. In this way, the amount of time that is necessary for the design process is reduced and designers gain more time for developing new collections;
- It also provides one with the opportunity of designing the garment models in a special CAD environment, which allows the personalization of the development process and the subsequent analysis. This enables the designer to find the best shape of the garment contour lines, which positively impacts the amount of material that is used (the goal is to design a model with minimal waste);
- The shape and the size of the model ensure physical comfort, so that the customer will use the purchased product for a long time. In this way, the life cycle of the product is extended, which is beneficial to the environment (it becomes waste only after long use).

The aim of this paper is to develop a solution for the ergonomic design of a jacket (business casual) based on a two-dimensional geometric pattern while integrating data regarding the dynamics of the human body to improve functionality and comfort.

With these aspects in mind, the paper is structured as follows: Section 1 presents a review of the literature on the application of ergonomic principles in the development of workwear and sportswear patterns; Section 2 presents the research design development process of personalised virtual prototypes using digital tools (Gemini's Made-to-Measure CAD system); Section 3 presents the analysis of the virtual prototypes of the model that are simulated on the customer's mannequin/avatar in static and dynamic positions by using Clo3D CAD; Section 4 presents the results of the developed solutions; Section 5 includes the final remarks on the results of the research, and Section 6 describes the main limitation and possible future developments in the field of ergonomic design of personalised clothing with different silhouettes and for various wearer categories.

## 2. Materials and Methods

### 2.1. Theoretical Consideration

#### 2.1.1. Conceptual Definition of the Jacket

Men's fashion has had to evolve significantly to keep up with the concept of the "modern man." Although it has changed in line with social challenges, the suit is always an option regardless of gender due to its formal function and aesthetic [12].

The business casual suit, which is a type of formal attire, emerged in the mid-19th century and quickly became popular among both the social elite and the working class. Its purpose is to make a good impression in professional settings. According to a market survey [13], in 2002, around 26% of men wore a business casual suit; in 2007, around 34% did so, and the percentages continue to change determined by each country's economic situation (formal business clothes represented 12% of men garments, both in 2002 and 2007). The Global Formal Wear Market size is expected to grow to USD 3.4 billion in 2021, representing a CAGR (Compound Annual Growth Rate) of 6% during the forecast period of 2021–2030. The demand for formal garment products has increased globally due to the rapid urbanisation and modernisation in developing countries. However, this growth is constrained by high prices and limited affordable options available in these regions [14].

In men's jackets, the shape of the collar, lapel, and sleeves; the shape and type of pockets; the closure system; the length of the shoulders and the product; and the garment fit are all essential details.

Formal clothing for men often focuses on proportions as the most important criterion for appreciation and, therefore, will influence the decision to purchase. In the basic sense of the word, "shape" refers to the fit of the garment on the customer's body (that is, the appropriate size for the customer), but the cut and style should also be suitable for the type and size of the wearer. There has been an increased demand for jackets with a higher degree of fit to the body, especially among younger buyers. The accepted standards of dress codes regarding appearance are always evolving, and these create a clear division between suitable products and those that are not.

The comfort of wearing a jacket is a result of the balance between the body, the environment, and the clothing, which is influenced by subjective sensations related to biophysical and physiological feedback of comfort or discomfort depending on the activities and health of the wearer.

An essential component of wearer comfort is the freedom of movement and breathing. The body dynamics while breathing and moving are influenced by the wearer's physical condition, age, gender, genetics, routine, type of daily activities, diet, geographical location, etc. Casual and leisure apparel must conform to these needs with minimal energy consumption of the muscles.

These elements of comfort can be achieved through a product's ergonomic design. However, since wearers' requirements can change over time; for example, current trends suggest they are opting for lighter products due to air-conditioned environments and climate change. The correct positioning of a pocket for easy access, a cut adapted to the type of activities most often performed, the choice of materials according to the intended use, full or partial lining—these ergonomic solutions and many more are available to fashion designers.

At the enterprise level, for personalization, data regarding the shape and dimensions of the human body are collected by either 3D scanning or by direct measurement of a static subject, depending on the equipment possessed by the company. The correctness of the anthropometric measurements, the choice of materials, the design pattern, and the choice of a suitable finishing technology are driven by consumer demands.

For fitted, multilayered clothing products (i.e., products that faithfully follow the lines of the body), designers must find a compromise between the customer's wishes and the possible shape of the model since a tight fit may restrict normal body movements and balance.

These requirements are the basis for the development of specifications in clothing manufacturing and are the starting point for product documentation.

The main objective of this case study was to create a virtual prototype that simultaneously met three consumer requirements, namely:

- Anthropometric;
- Hygienic, and;
- Psycho-physiological.

The wearer's comfort and ease of movement was limited by the fit of the analyzed jacket.

2.1.2. Characteristic Elements of the Dynamics of the Human Body

An essential requirement for clothing is comfort. For clothing products with a high degree of adaptation, comfort is often related to ease of movement in daily activities (e.g., mobility of the boot and upper limbs, breathing, etc.). Garment comfort is achieved through ergonomic design that incorporates information regarding body dynamics.

These products must have differently shaped cutouts for the neck and arms. The shape of the product should consider the shoulders of the wearer, which is mainly determined by the anatomical features of the shoulders, the posture, and the chest. The trunk must align and stabilize the spinal segments, both during simple vertical standing and movement.

The mobility of the spine results in trunk movements in three planes, namely [15]:

1.  In the sagittal plane, with flexion forward and extension backwards;
2.  In the frontal plane, lateral inclinations to each side, and;

3. In the transverse plane, with rotation.

Body movement involves three musculoskeletal systems: the skeletal system, the muscular system, and the articular system. The movements of the body can be performed in a variety of directions and involve many joints. Chest movements are related to the act of breathing. Muscle contractions affect the mobility of various segments. In the movements of the thorax, a rhythmic sequence can be observed, expansion corresponding to inhalation and contraction corresponding to exhalation. Therefore, the thoracic cavity must have a certain mobility to allow for the movements involved in breathing. The upper limbs perform abduction and adduction movements. At the level of the shoulder joint, complex movements can be performed in several planes simultaneously, and at the elbow joint level, movements are performed in two planes [15].

The trunk and upper limbs determine the changes in body dimensions. These changes are influenced by age, daily activities, professional activities, lifestyle, diet, and the type of movement and amplitude.

The dimensional characterization of the human body has been determined through a series of indicators and anthropometric measurements. To obtain their values differentiated by age and gender, anthropometric studies on a sample representative of the intended use have been conducted.

The human body dimensions measured in the static position were static dimensional characteristics and denoted by $X_i$ ($^s$). The same dimensions measured in different dynamic body positions were dynamic dimensional characteristics and denoted by $X_i$ ($^d$). The difference between the value of a dimension measured in a particular dynamic position versus the static position was the "dynamic effect" and was expressed in either absolute values, d (cm), or in relative values, $e_d$ (%) [16,17].

For each anthropometric dimension, the dynamic effect was calculated using the relationship:

$$d_{ij} = X_{ij}(^d) - X_{ij}(^s), \tag{1}$$

$$ed_{ij} = \frac{x_{ij}^{(d)} - x_{ij}^{(s)}}{x_{ij}^{(s)}} 100. \tag{2}$$

The abbreviations used have the following meanings: $d_{ij}$ and $ed_{ij}$ represents the absolute and relative dynamic effect, dimension "i" for the subject "j"; $X_{ij}$ ($^s$) represents dimension "i" of the subject "j" measured in the static position; $X_{ij}$ ($^d$) represents dimension "i" of the subject "j" measured in a particular dynamic position.

The data concerning dynamic effects was essential, but their use in the construction of patterns (partial or complete) would need to be differentiated according to the nature of the product and its intended use.

### 2.2. Designing the Geometric Shapes of a Fitted Jacket with Dynamic Effect Data of the Torso

Product design using ergonomics considers the garment's intended use and destination and ensures a high degree of agreement between the requirements and the implementation. A detailed analysis for ergonomic design should include all the requirements for the properties that define compliance with the anatomical, physiological, and psychological characteristics of the body and ensure comfort and safety when worn to alleviate physical and psychological stress; these requirements should be based on the ergonomic properties of the system, "body–clothing–environment".

Once established, design criteria should be used to critically analyze existing products for improvement [18,19]. We should consider the ergonomics of the product as the specific form (or field) of ergonomics to collect and use empirical data (e.g., psychological, medical, technical, economic, etc.) for the adaptation of products for human needs (e.g., psychophysiological, shape, size, etc.) while increasing the comfort and the effectiveness of the product for its intended use.

The customization of personalized clothing products requires the use of software (e.g., Lectra, Gemini, Clo3D, Grafis, etc.) that integrates the information provided by the customer to obtain a virtual prototype that can be viewed in 3D.

The design software, after fitting the virtual prototype to an avatar with the dimensions and the shape of the customer, analyzes the values between the customized prototype and the technical sketch and the balance of the product as projected on the virtual mannequin (i.e., avatar). When these requirements are reconciled, the ergonomic design requirements of the product are determined as follows:

- If the requirements are met, the applied design solution is validated, and the samples of the product elements are transformed into production patterns to be used in the manufacturing process;
- If certain design deficiencies are identified, the design software corrects these deficiencies and continues the simulation process to revalidate the model.

The design process of personalized virtual prototypes in this study performed the following steps for a personalized, business-style men's jacket:

(a)  Designed the shapes of the elements/parts based on the data obtained from the static body position that matched the product data;

(b)  Analyzed the virtual prototype of the model on the mannequin/avatar of the customer in static and dynamic positions;

(c)  Determined the values for the dynamic effects to be applied to the ergonomic design of the product elements;

(d)  Analyzed the new virtual prototype in static positions as well as in frequent dynamic positions.

## 3. Results

*3.1. The Design of the Shapes of the Elements/Parts Based on the Information concerning the Human Body (Measured in the Static Position), Confirmed by the Data Provided by the Product*

Business casual attire is the new dress code in the international professional environment, accepted by those in senior management. It consists of: a fitted jacket, a pair of fitted trousers, a fitted shirt, a pair of classic shoes (Oxford or Derby style), a handkerchief for the top pocket or tie, a pair of socks, and a classic bag. This type of jacket is very popular because it can be manufactured in a wide range of colours and its stylistic elements (the shape and the size of the collar of the lapels, the buttoning system, the shape of the edges, the presence of slits at the ends of the sleeves, the types of pockets and the finishing method that has been used for them, etc.) can be customized in many ways.

The jacket model chosen for this case study (Figure 1) was suitable as a current business outfit. It had a high fit, a classic cut, and two-piece sleeves. The product closed with a single row of buttons, and the top had a collar and lapels. The product had a pocket with a loop/chest pocket (on the top of the left side) and two pockets with flaps below the waist.

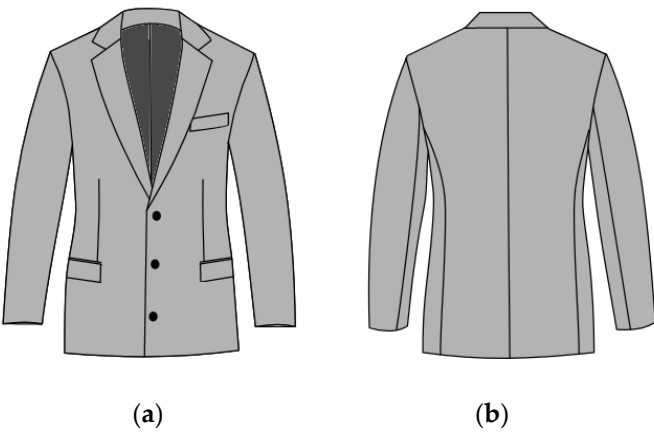

(**a**)  (**b**)

**Figure 1.** Jacket model: (**a**) Front; (**b**) back.

The adjusted silhouette makes it more difficult to ensure freedom of movement, especially in the dynamics of the wearer's back side of the torso when they raise their arms. The dynamics of the body also make it more challenging to ensure a balanced position of the product (shifting the shoulder line, changing the position of the collar and sliding the product backwards, moving the position of the edges and sleeves), adds mechanical stress on some attaching lines (back of the attaching line between the sleeves and the main component of the product, the line in the middle of the back, the outer seam of the sleeve), and makes it more difficult to design a product of esthetical value.

When moving the upper limbs or torso, certain dimensions body (especially those on the back of the torso) suffer significant changes in their values relatively to those measured in the orthostatic and relaxed position of the subject. Information on the dynamics of the human body is used in the design of personal protection products and sports gear, but they have not been integrated into the design stages of fashion products, especially for multilayered ones, due to the high diversity of models, materials and manufacturing process, the various types of stress that develop between the layers of material when the body makes a certain movement, and because the number of layers of material differs on product areas and the geometries of the contour lines are complex.

The functionality of the jacket, especially the adjusted one is designed, in this sense, The authors of this paper have designed an adjusted jacket by taking into account the parameters describing the dynamics of the human body, which have been calculated by using the values of the anthropometric indicators when the body was in different static and dynamic positions in order to optimize the shape of the patterns (ergonomic design).

The design of the product elements was conducted according to a geometric method using specific functions of the Gemini CAD system and the Made-to-Measure module [20]. The MTM module processes mathematical relations using specific functions of the CAD environment and automatically resizes the shape of the parts based on updates to the initial data.

The mathematical relations necessary to obtain the geometry of the product parts [21] were adapted to the selected jacket. The silhouette of the product was determined by the value of the following allowances: chest, waist, and hips. The allowance's value at the chest (which was considered the basic allowance) highly influenced the silhouette of the product; therefore, depending on its value, the values of the waist and hips were adjusted. Depending on the silhouette of the product, the structure of the layers, and the characteristics of the customer's physique and posture, the value of the chest allowance was distributed among three constructive areas of the product: front, side and back.

Figure 2 shows the design sequences of the 2D shapes of the model parts in the geometric plane. Changing the structure of mathematical relations, the values of initial data, or required categories of information determined the automatic recalculation of the shape of the parts connected by their main points to the geometric layer of the design. Two-dimensional shapes were used to obtain the 3D virtual prototype [18].

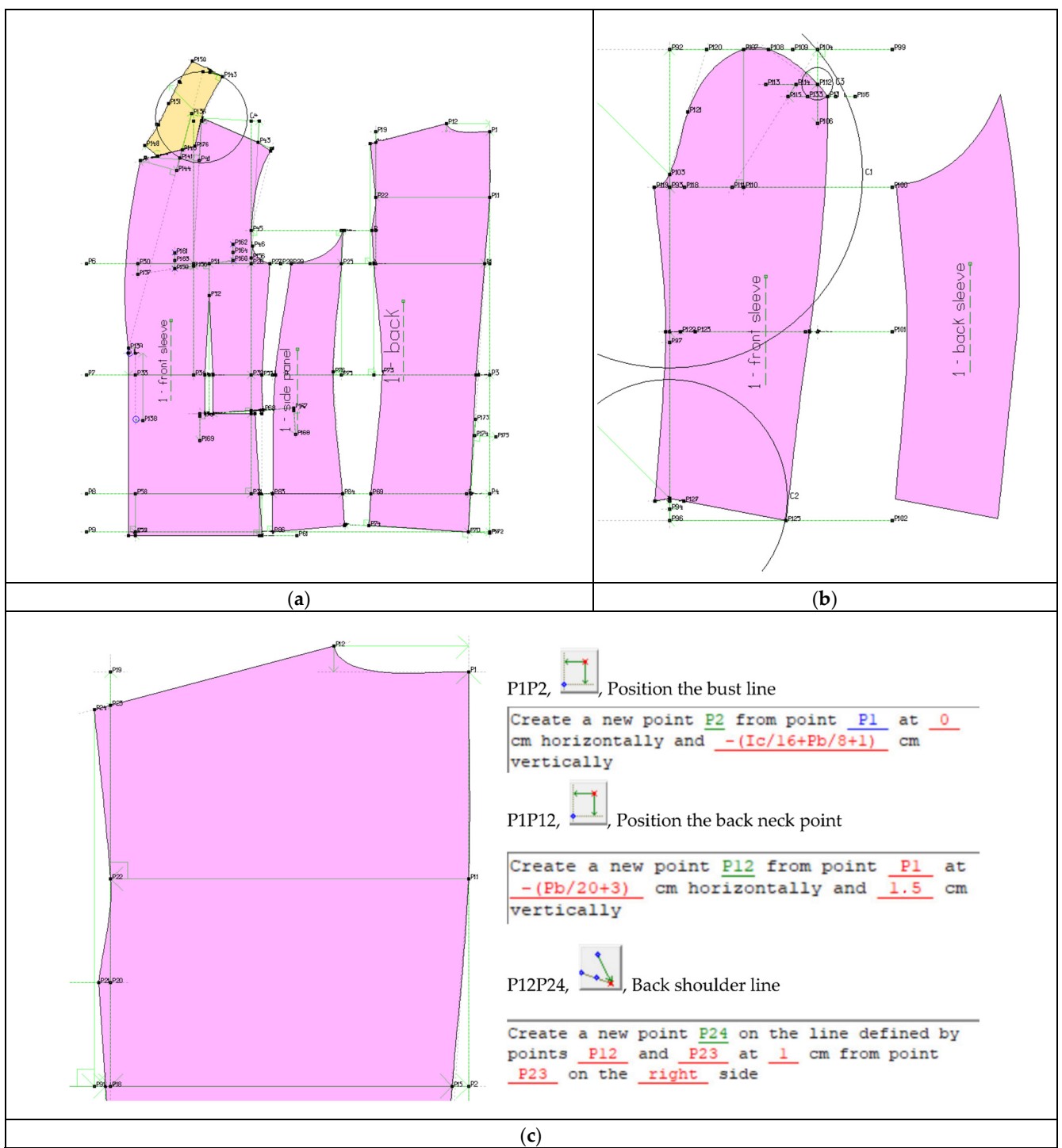

**Figure 2.** *Cont.*

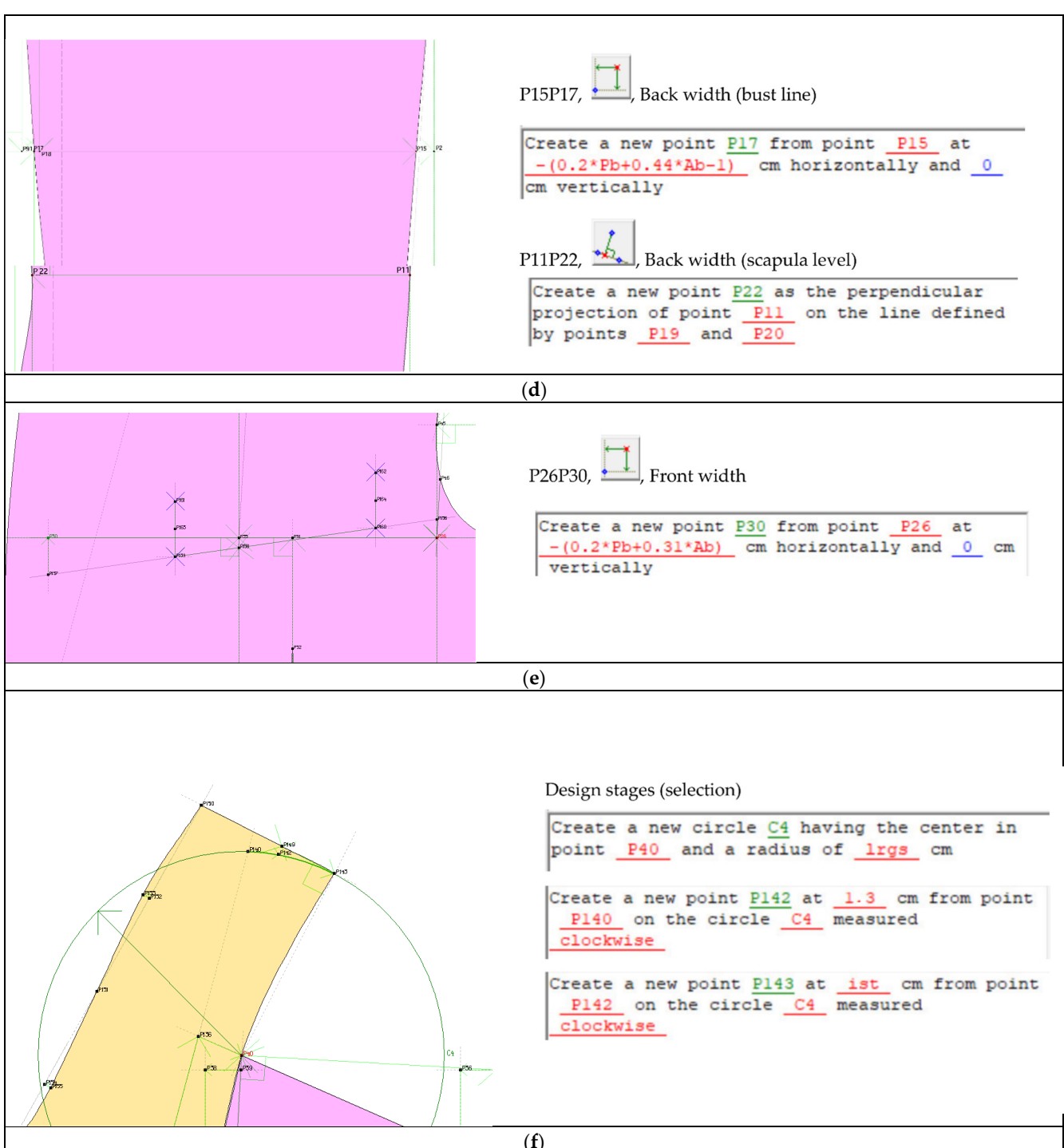

**Figure 2.** *Cont.*

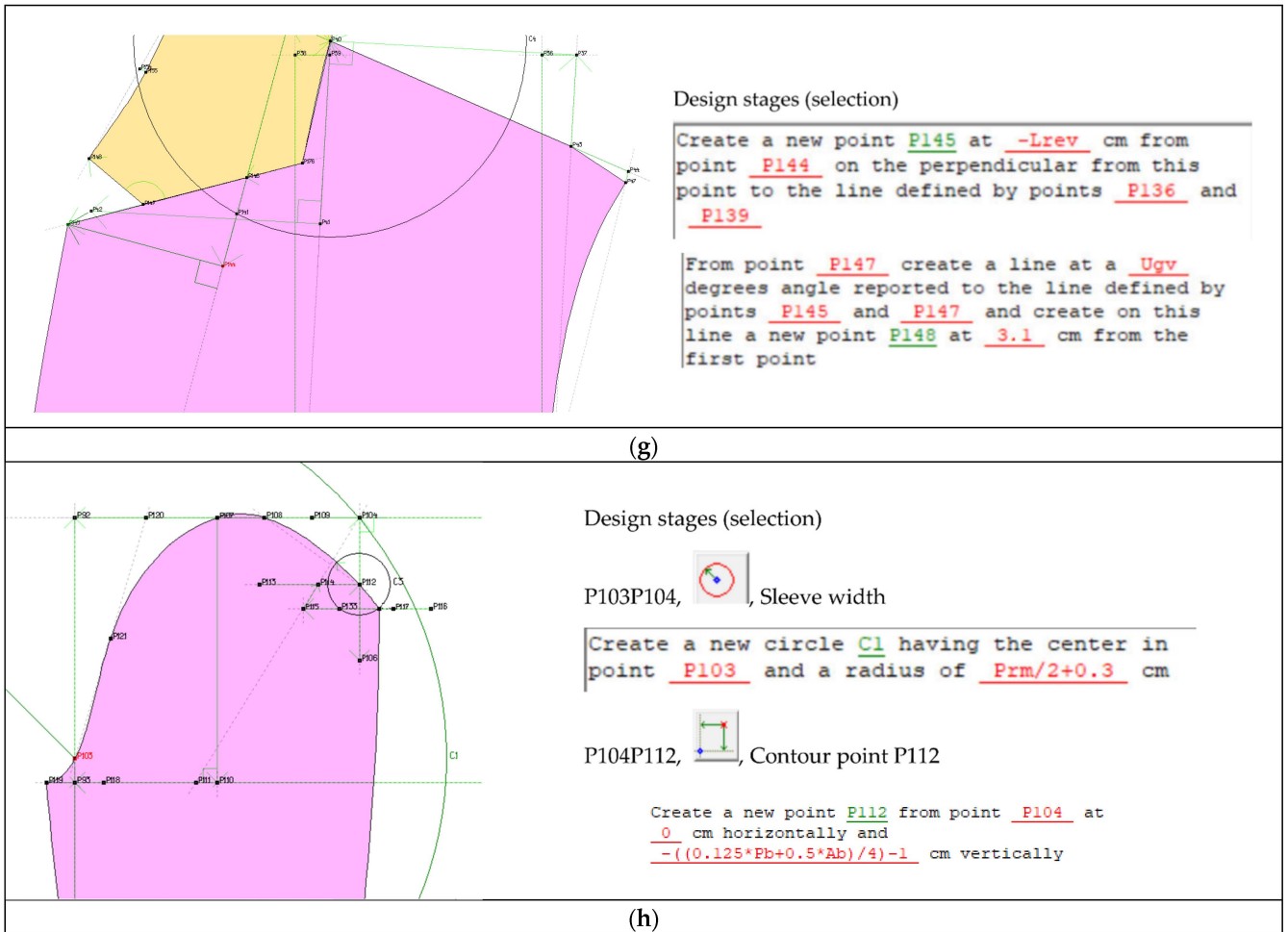

**Figure 2.** Two-dimensional shapes of the model parts imported in 3D space to obtain the virtual prototype: (**a**) Jacket model (front, side panel and back); (**b**) jacket model (sleeves); (**c**) back pattern (upper part); (**d**) back pattern (upper part); (**e**) front pattern; (**f**) collar pattern; (**g**) lapel pattern; (**h**) front sleeve pattern.

*3.2. The Analysis of the Virtual Prototype of the Model Dressed on the Mannequin/Avatar of the Customer in Static and Dynamic Positions*

For this case study, the balance and fit of the product on the avatar were analyzed in static and dynamic positions to determine how well the product fit on the different parts of the body, whether the constructive, stylistic details of the model were preserved, and whether there were areas of the product with possible constructive defects (e.g., free, fixed, or oblique folds). When examining the folds, there were issues with sizing and mismatched parts [22].

At this stage, it was necessary to use a custom avatar that reflected the shape of a customer's body. The dimensions and the shape of the avatar were obtained by changing the dimensions of a virtual body in the database of CAD software (Clo3D) [23]. The working method included the following steps: enter the values of anthropometric measurements (of the client), choose the position of the avatar (suitable for 3D simulation), and import the parts of the model designed in the geometric plane (personalized). The avatar had free upper limbs next to the torso in the orthostatic position. The values for the positioning angles of the upper limbs were α = 8.23° and β = 167.23° (Figure 3).

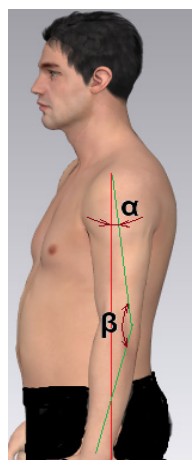

**Figure 3.** The angles of the upper limbs.

When creating the virtual prototype, the typical layers for the jacket were considered: base material, lining and interlining, the relative position of the product on the body, and undergarments. Visualization of the virtual prototype on the customer's avatar included multiple viewing angles, and the balance of the product was checked on the avatar. The virtual model was rendered for this stage (Figures 4 and 5).

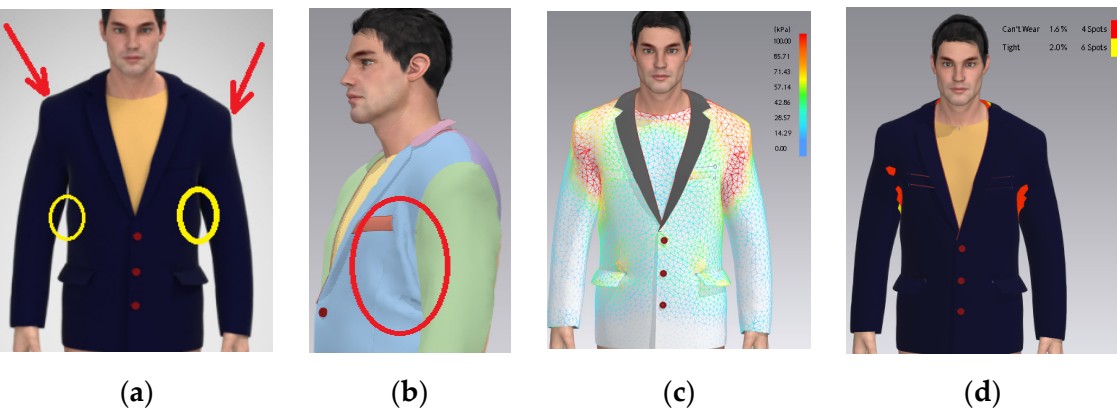

(**a**)  (**b**)  (**c**)  (**d**)

**Figure 4.** Jacket model (front details): (**a**) rendering process; (**b**) obliques free folds; (**c**) stress map; (**d**) fit map. The yellow and the red circles represent folds.

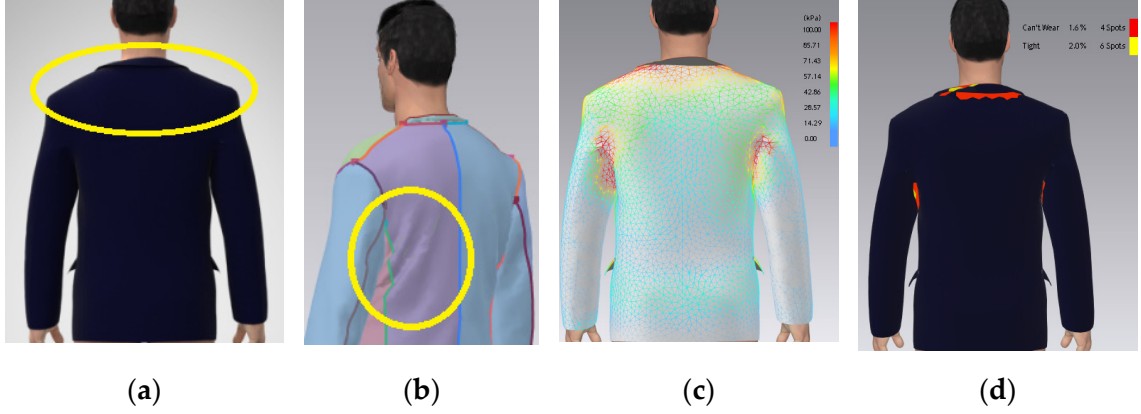

(**a**)  (**b**)  (**c**)  (**d**)

**Figure 5.** Jacket model (back details): (**a**) rendering process; (**b**) obliques free folds; (**c**) stress map; (**d**) fit map. The yellow circles represent folds.

From the analysis of the presented images (Figures 4 and 5), the following conclusions were drawn:

- The model did not have the correct position at the level of the shoulder line (the length of the shoulders was too long); for this reason, there were slips and tension points on the back (Figure 5a,b);
- The distribution of the chest allowance did not provide a comfortable width at the side (Figures 4c and 5c);
- The sleeve had not been properly placed; the positioning of the sleeve was due to the sizing of the upper section. (The pictures indicate tension zones in the upper section of the sleeve determined by the sliding slope, Figure 4c);
- The width of the sleeve at the elbow and at the finishing line allowed for easy placement;
- The presence of oblique, free folds near the side (Figures 4b and 5b); this indicated incorrect sizing for the customer;
- There were no fit issues at the waist or hips level;
- Depending on the physical and mechanical properties of the materials, the silhouette of the model and the shape of the customer's body caused tension in the layers of the jacket (Figures 4c and 5c). This is symbolized by the following colours:
  - The blue colour indicates normal tension (the product fits perfectly);
  - The green colour indicates a lack of tension expressed by excessive lightness;
  - The red colour indicates a high-tension value (the product is excessively tightened around the body and would not be comfortable to wear).

When interpreting the data provided by the stress map (Figures 4c and 5c), the stress areas were placed as follows: the upper back and front, the upper contour of the side panel, and the upper section of the sleeves.

When analyzing the fit of the product on the body (Figures 4d and 5d), the problem areas were the same as previously indicated. Under real-world conditions, a wearer would feel discomfort in these areas.

The process of simulating a virtual prototype for fashion products has often been performed on the avatar in a static position. However, clothing for competitive sports and protective products (e.g., activities in which the human body performs repetitive movements at a certain amplitude), the simulation of the fit of the product is performed on dynamic, even extreme, positions, as dictated by the nature of the activity.

The simulation process of the model (Figure 1) was performed in static positions and for typical dynamic positions of the trunk and upper limbs. The dynamics of a clothed body include the trunk's forward tilt, with/without forward movement of the upper limbs, and the act of breathing. The average torso tilt angle (i.e., normal torso dynamics) was $9.5° \pm 2.9°$ (a value determined by gender, age, daily activities, lifestyle, diet, etc.) [15].

The dynamic positions have been simulated in the virtual environment by using the modified virtual mannequin (bent) that represents the customer's body. The static pose was modified as follows: The trunk was bent (Figure 6a) forward through the lumbar joint of the spine, and the upper limbs were directed forward through the shoulder joint. For this position of the avatar (inclined torso and upper limbs in horizontal alignment), the 3D simulation of the projected model was performed. The dynamic pose of the 3D simulation of the selected model is shown in Figure 6.

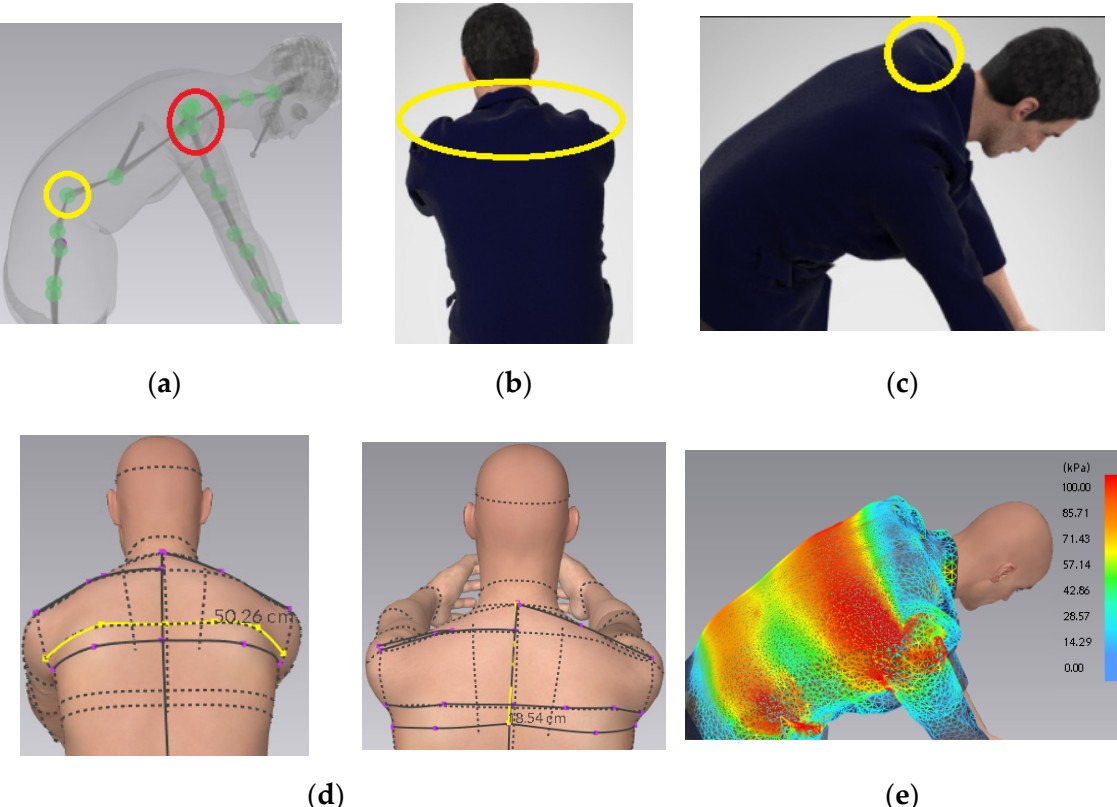

**Figure 6.** Dynamic pose: (**a**) bent trunk; (**b**) jacket (back view); (**c**) jacket view (lateral view); (**d**) body measurements (dynamic position); (**e**) jacket (stress map). (**a**) The yellow and the red circles highlight some joints. (**b,c**) The yellow circles represent folds on the upper part of the backside.

The following was found:

- The presence of transversal, free folds near the collar (Figure 6b,c); this indicated incorrect sizing for the customer (back width);
- In this position, the width of the back (torso) changed considerably (e.g., the width of the back measured at the projection of the scapula, the width of the back measured between the posterior axillary points, the length of the back from the cervical vertebrae to the axillary point), see Figure 6d;
- The stress map (Figure 6e) showed that the dimensioning in the transverse direction was aggravated in the upper section of the product. This finding indicated the need for an additional analysis of the human body dynamics and the subsequent integration of this information for the ergonomic design of the product elements.

*3.3. Determining the Values of the Dynamic Effects of the Anthropometric Measurements and the Ergonomic Design of the Shape of the Product Elements by Integrating the Values of the Dynamic Effects*

For this category, the support surface was defined by the contour of the base of the neck, the line of the shoulders (left/right), by the curvature of the sternum, and by the curvature of the shoulder blades.

Typical body dynamics included the act of breathing, the forward movement of the upper body, and the movement of the upper limbs. Under these conditions, the body measurements that registered significant changes in values, as compared to the orthostatic position, were measured at the upper part of the trunk.

The anthropometric quantities selected were (see Figure 7):

(1) Length measured from cervical vertebral point to axillary vertebral point (ARS);
(2) Length from cervical vertebral point to waist ($L_t$);
(3) Circumference of the chest at deep inspiration ($P_b$);

(4)    The width of the back was measured at the level of the posterior axillae ($\ell_s$).

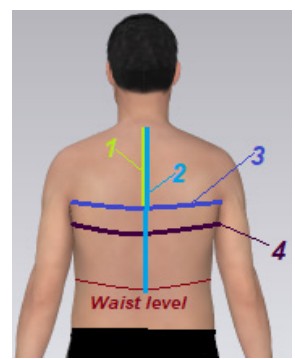 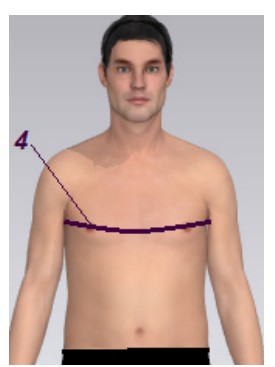 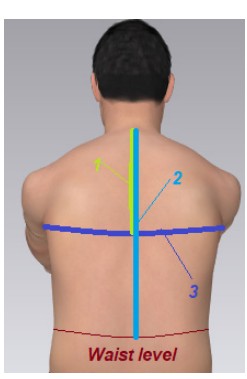

**Figure 7.** Anthropometric quantities.

The absolute values of the dynamic effects were determined on a sample of 50 male subjects aged 34–60 years old with occupational activity that did not require intense physical effort. The measurements were performed directly (Figure 7). They were taken using the direct method (anthropometric measurements were made on the subject's body).

Based on the recorded data, the absolute values of the dynamic effects were determined with relationship (1).

The individual values of the selected body sizes were processed using statistical and mathematical methods. The values of the following statistical parameters were determined: minimum value ($X_{min}$), maximum value ($X_{max}$), average value ($\overline{X}$), mean square deviation ($S_{\overline{x}}$), coefficient of variation ($C_v$), and test value of the selection average ($T_{\overline{x}}$).

The selected confidence level of statistical processing was 95%. It was concluded that the results obtained from statistical processing were significant and could be used to characterize the phenomenon for further research if the value was greater than the value of the Student's *t*-test ($T_S = 2.009$).

From the analysis of the data summarized in Table 1, the following conclusions were drawn:

- The results obtained were statistically significant ($T_{\overline{x}} > T_S$); these data expressed a real phenomenon and could be used to characterize it;
- The values of the coefficient of variation ($C_v$) expressed the degree of dispersion/grouping of the individual values with the average value. In anthropometric research [16], the following intervals have been defined: (0; 10%) → a high degree of homogeneity, i.e., the individual values of the analyzed dimensions are grouped among themselves and around the average value; (10%; 20%) → indicates a medium homogeneity degree; (20%; 30%) → a low degree of homogeneity for the values of the analyzed quantity.
  - ✓ The values of the coefficient of variation $C_v$ for the chest circumference (static, dynamic, and dynamic effect) were in the first interval, which indicated a high homogeneity (the individual values were close to each other and at the average value). The same degree of homogeneity was found for $L_t$ in the static position.
  - ✓ For ARS (static, dynamic, dynamic effect), $L_t$ (dynamic and dynamic effect), and $\ell s$ (static), the degree of homogeneity was medium, and the values were relatively close to the average value, but for $\ell s$ (dynamic and dynamic effect), the homogeneity was low, indicating that the individual values were far from the average value and from each other.
  - ✓ The width of the back was the anthropometric dimension that had a very high value for $C_v$ (dynamic and dynamic effect). In highly customized products, the value of the dynamic effect had to be integrated into the design process of the product elements to ensure free movement (the value of the dynamic effect was fully or partially integrated, depending on the value of the constructive allowance);

✓　The values of the dynamic effect $L_t$ had a medium dispersion (medium homogeneity, close to the upper limit of the range). The average value could be used in the design process of the pieces of the product but had to be supplemented by the value of a constructive allowance in the longitudinal direction. The textile used to manufacture the product had medium elasticity, so this material property partially compensated for the dynamic effect of $L_t$ and provided wearing comfort.

✓　The values of the coefficient of variation for ARS were higher than for $L_t$; the movements of the trunk and the upper limb determined this degree of variability. The values obtained for the dynamic effect were compensated by adjusting the dimensions of the construction segments in their direction of measurement.

**Table 1.** The values of statistical parameters.

| Statistical Param./ Body Dimension | | $X_{min}$ (cm) | $X_{max}$ (cm) | $\overline{X}$ (cm) | $S_{\overline{x}}$ (cm) | $C_v$ (%) | $T_{\overline{x}}$ |
|---|---|---|---|---|---|---|---|
| ARS | Static | 15 | 29.5 | 20.4 | 2.78 | 13.6 | 7.34 |
| | Dynamic | 16.1 | 31.1 | 21.79 | 2.76 | 12.66 | 7.89 |
| | Dynamic effect | 1.1 | 1.6 | 1.37 | 0.17 | 12.36 | 8.08 |
| $L_t$ | Static | 39 | 61.2 | 49.5 | 4.21 | 8.50 | 11.7 |
| | Dynamic | 39.5 | 78 | 56.3 | 6.07 | 10.77 | 9.28 |
| | Dynamic effect | −14 | 23 | 6.83 | 4.65 | 19.78 | 3.48 |
| $\ell_s$ | Static | 35 | 59 | 43.9 | 4.46 | 10.14 | 9.86 |
| | Dynamic | 43 | 73 | 57.02 | 5.63 | 29.87 | 10.12 |
| | Dynamic effect | 1 | 27 | 2.26 | 0.65 | 28.7 | 3.47 |
| $P_b$ | Static | 78 | 127 | 100.65 | 9.25 | 9.18 | 10.88 |
| | Dynamic | 81 | 131 | 104.14 | 8.92 | 8.56 | 11.67 |
| | Dynamic effect | −2.3 | 3.5 | 3.48 | 3.24 | 9.29 | 10.76 |

The value of the constructive allowance at the chest line, which was used in the sizing of the jacket, had a lightness allowance. The value of this allowance was calculated based on the value of the dynamic effect for $P_b$ and the relative position of the product on the body. Studies in the field of dynamic anthropometry [16] have shown that the change in chest circumference in adult men during breathing is 2.5 cm. In the case of the jacket, this required minimum deviation was 2.5 cm; if the design deviation at the chest line had a value greater than 2.5 cm, the designed product met the requirements to ensure breathing.

●　The values of the mean square deviation confirmed the conclusions based on the $C_v$ values.

Based on the aforementioned conclusions, the shapes of the model parts were redesigned. In the dimensioning of the constructive segments specific to the product elements, the conclusions derived from the dynamic analysis of the human body were integrated to ensure the design of the shapes according to ergonomic criteria.

The following changes were made in the ergonomic design of the model parts:

(1)　Changed the value of the base allowance (the chest line) to acquire the model silhouette. Several values for this allowance were tested in the range of 5–9 cm, as recommended in the literature [16]. For each value, the simulation of the 3D virtual prototype on the avatar was performed (static conditions), and the placement of the product on the avatar was analyzed;

(2)　The assumption of a different distribution of the basic allowance (front, side, and back) than the one originally used (for the allowances used in point 1) was established;

(3)　The value of the allowance was changed longitudinally (back height); values in the range of 3.2–3.6 cm were tested in correlation with the values of the chest allowance and the body shape;

(4) Part of the value of the dynamic effect for ℓs and ARS was contained in the cross-sectional design of the shape of the product elements. The full value of the dynamic effect was not used for the following reasons: the designed model had a single-breasted closure with a long lapel that determined a certain lightness at the chest line. If the full value of the dynamic effect had been integrated, it would have increased the surface area of the product element and the product would not have fit properly on the body. In addition, the textile materials that would be used in the manufacturing process were characterized by elasticity, a property that promotes comfort;

(5) We changed the width of the lapels; as a result, the suggested appearance of the model was achieved;

(6) We changed the size of the sleeve pattern (elbow and end line). The width of the sleeve at the depth line changed automatically: the circumference of the sleeve armhole changed with the change in the mathematical relations of the dimensioning of the product elements. The original shape of the sleeve was light compared to the customer's preference.

For this case study, the changes in Made-to-Measure (Gemini CAD) were located in the geometric layer, where the design pattern was elaborated. The new shapes of the product elements were generated automatically as the main points on their contour are magnetized by the points of the geometric plane (Figure 8).

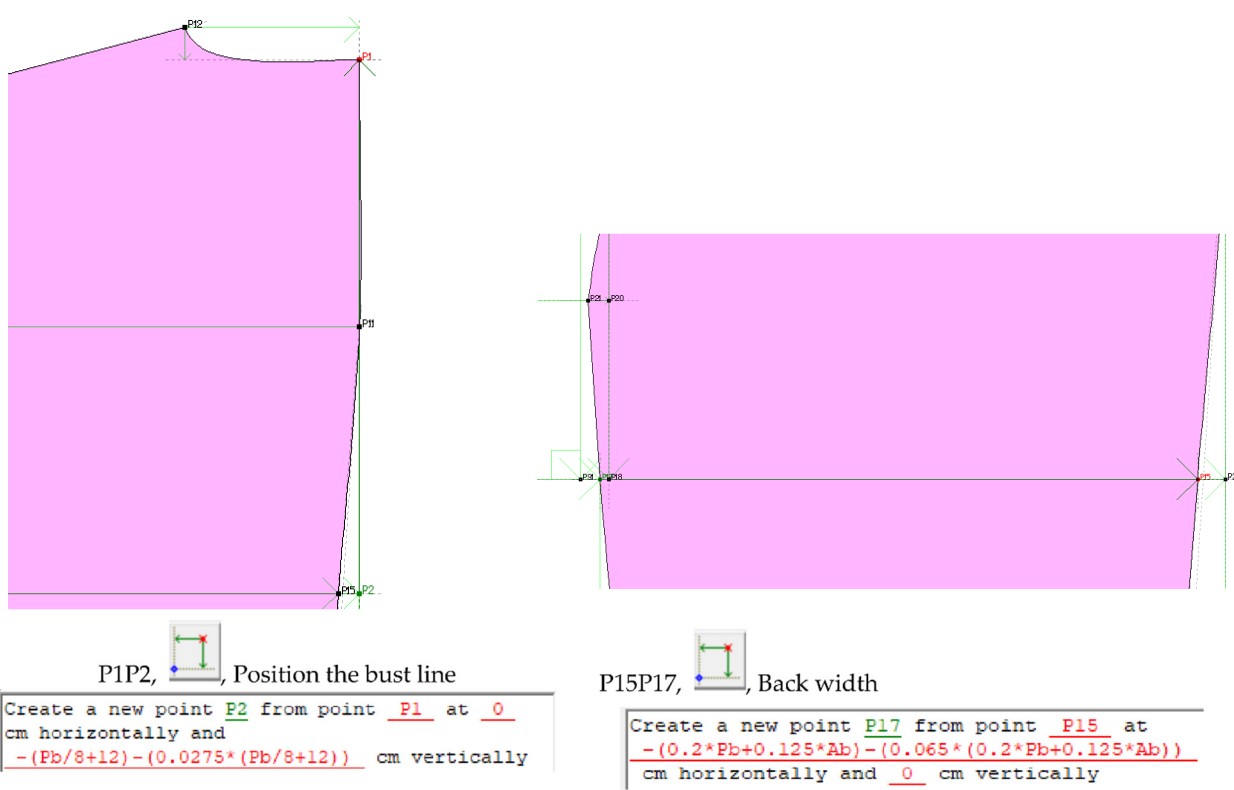

**Figure 8.** *Cont*.

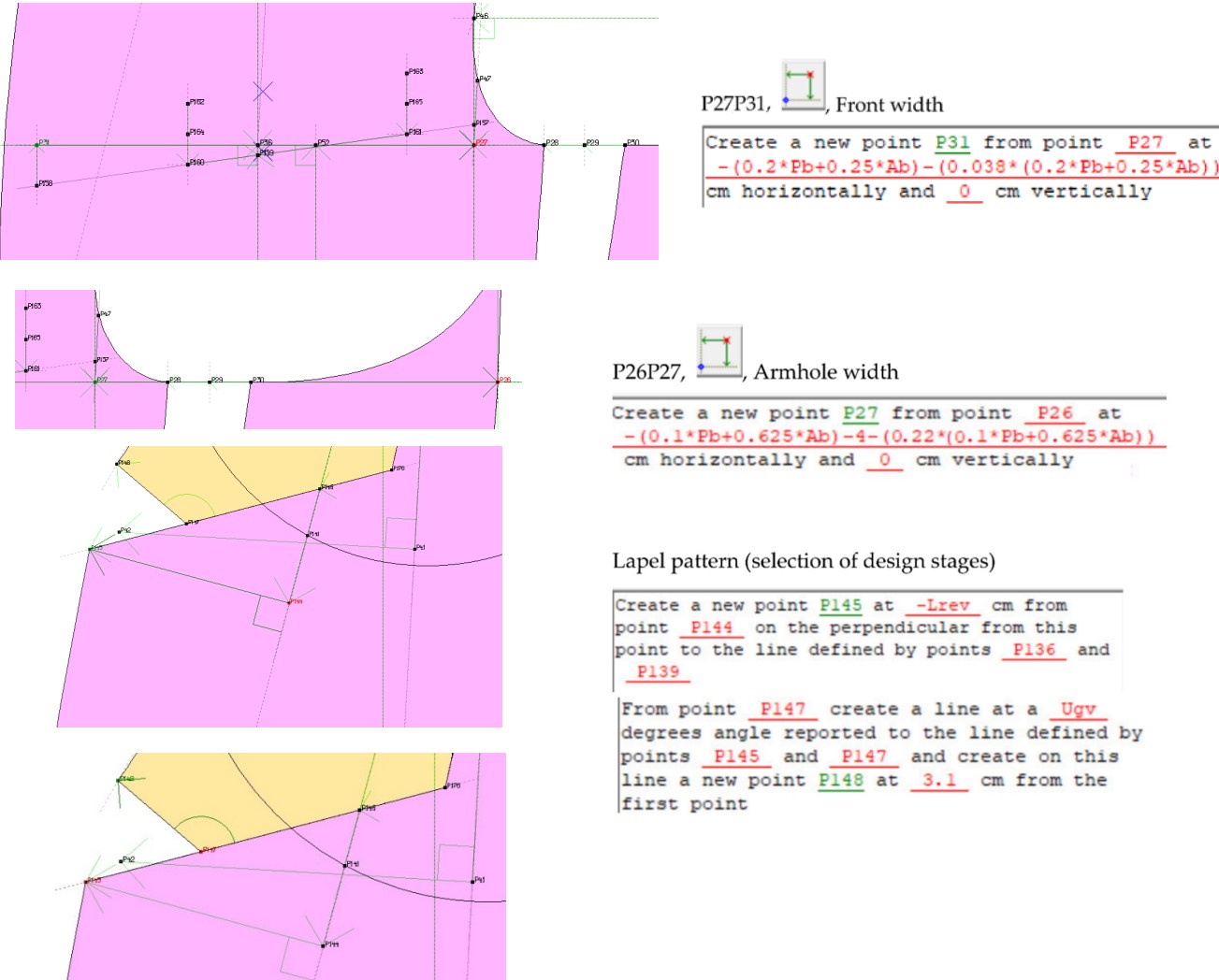

**Figure 8.** Jacket re-design stages (selection).

Designing according to the ergonomic criteria of the shape of the elements of clothing products (fashion) had the following objectives:

- Optimize the 2D shapes of the elements in the design and the 3D volume of the prototype (in the first phase, it was virtual) following the requirements of the model and the body shape of the customer;
- Obtain a virtual prototype with a balanced placement on the customer's virtual body;
- Ensure the parameters for wearing comfort;
- Satisfy the customer's requirements for a clothing product that was both useful and ensured their psychological wellbeing.

After the necessary changes had been made, the model was saved and ready for 3D simulation. The simulation was performed under static conditions in successive steps according to points 1–6 to determine whether the changes had led to the desired result. Once the virtual model had been validated (static condition), it was tested for normal body dynamics.

### 3.4. The Analysis of the New Virtual Prototype (for Static Conditions and Frequent Dynamic Conditions)

The position of the product on the body was evaluated at different angles along with the following parameters: the sleeve angle position, as compared to the body of the product

($\alpha'$, expressed in degrees, Figure 9) and the tension (stress) in the back upper section of the product (P, expressed in kPa).

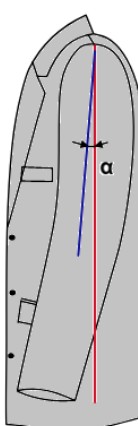

**Figure 9.** The sleeve angle position. The purpose of the colored lines is to illustrate the sleeve angle.

The relationship was evaluated between the values of the constructive allowance $A_b$ (cm) and the longitudinal allowance distributed along the back height ($A_{ars}$, cm), as independent variables (Xi), as well as the values of the sleeve angle ($\alpha'$) and the stress in the upper back section, as dependent variables (Yi). To evaluate the conformity of the product on the body surface, $A_b$ and $A_{ars}$ were chosen as independent variables since these variables had the greatest percentage of influence on the shape and the surface of the product sections, respectively.

We tested 10 combinations of values for $A_b$ and $A_{ars}$ for the same customer avatar.

The test to determine the relationships between the selected parameters was performed using the program Table Curve [24].

Different mathematical models were tested: simple, polynomial, rational, and non-linear. The adequacy of the models was tested using the Fischer–Snedecor distribution, and the significance of the coefficients of the mathematical model was tested using Student's *t*-test.

Based on the analysis of the diagrams shown in Figures 10 and 11, the following conclusions were drawn:

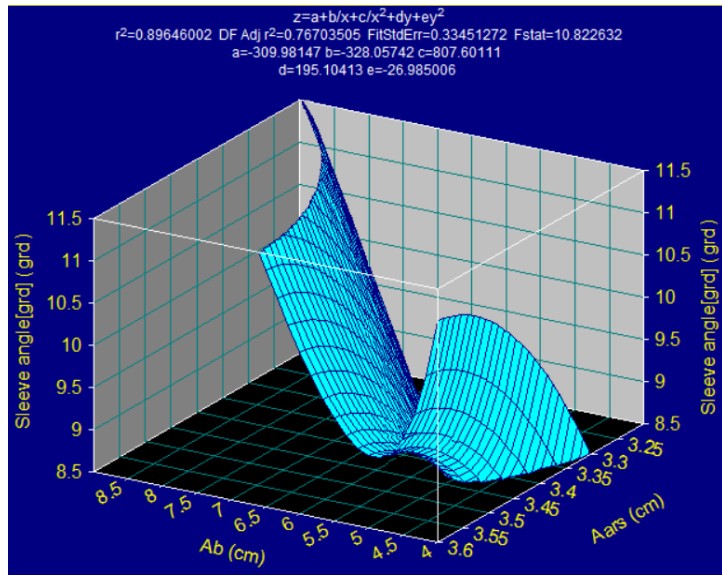

**Figure 10.** Correlation between the sleeve angle ($\alpha'$) with $A_b$ and $A_{ars}$.

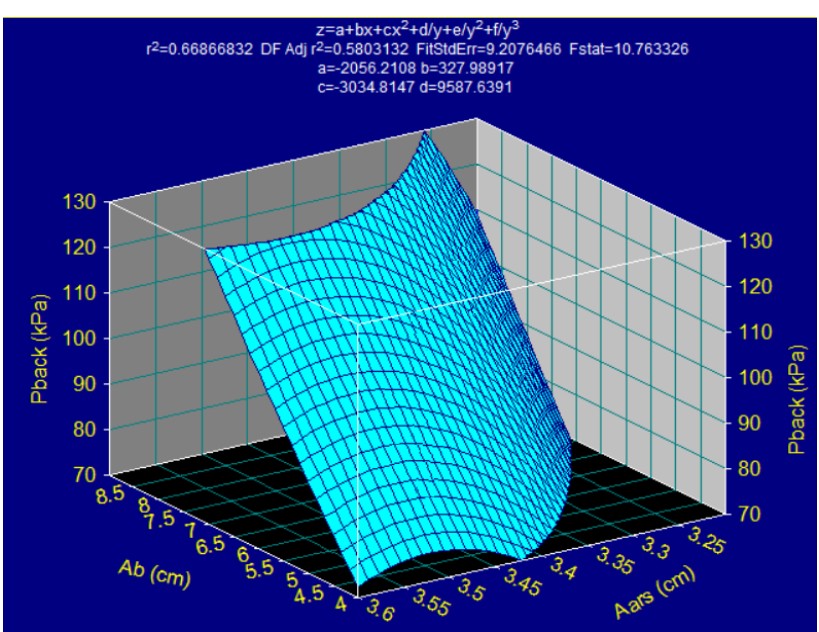

**Figure 11.** Correlation between the back pressure ($P_{back}$) with $A_b$ and $A_{ars}$.

- There was a strong correlation between the selected variables. The value of the multiple correlation coefficient $R_{y,Xi} = \sqrt{r^2}$ ($r^2$ was the coefficient of determination) was high (Figures 9 and 10). In anthropometric research, a multiple correlation test $R_{y,Xi}$ value in the range of 0.75–0.99 indicates a strong correlation.
- The correlation was not random, as the value of the coefficient of determination $r^2$ was greater than its error.
- For a value of the $A_b$ allowance in the range of 4–5 cm or a value of the $A_{ars}$ allowance of 3.3–3.6 cm, the positioning angle of the sleeve with respect to the body of the product had the value closest to the positioning value of the upper limb with respect to the trunk (orthostatic position). The value of the upper limb angle was 8.23°, and that of the sleeve was 9.25°.
- In Figures 3c and 4c, tension value (stress) in the range of 0–25.5 kPa indicated the absence of pressure (high ease); a value in the range of 25.5–42.5 kPa indicated the presence of pressure; a value in the range of 42.6–58 kPa indicated normal pressure (comfort); a value in the range of 58.1–71.5 kPa indicated the presence of pressure that would cause minimal discomfort; a value in the range of 71.6–86.5 kPa indicated the presence of pressure that would cause moderate discomfort; and a value in the range of 86.6–100 kPa indicated the presence of pressure that would restrict movement and cause severe discomfort. The graphs in Figures 10 and 11 show that the top of the product fell in the range where a customer would feel comfortable with a value of $A_b$ = 4–4.5 cm and $A_{ars}$ = 3.42–3.6 cm.

Figures 10–12 indicate that the product had a balanced position for $A_b$ = 4–4.5 cm and $A_{ars}$ = 3.42–3.6 cm, which would ensure comfort. However, there were still comfort issues in the front, specifically in the upper section of the side panel and the front sleeve. Relatively high pressure was eliminated by resizing and reshaping the elements of the product, as shown in Figure 12.

After the changes had been made and the new prototype was produced, it was tested in a dynamic position to verify the agreement between the body and the product (see Figure 13).

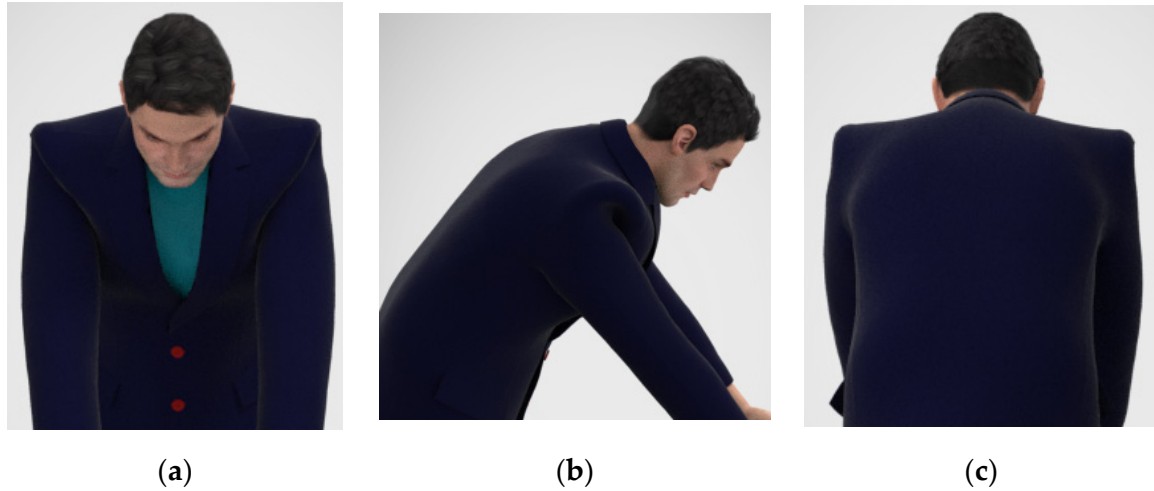

**Figure 12.** The new shape of the jacket: (**a**) Jacket model (front)—(**1**), the result of the rendering process; (**2**) stress map; (**3**) fit map; (**b**) jacket model (back)—(**1**), the result of the rendering process; (**2**) stress map; (**3**) fit map.

**Figure 13.** Final shape of the jacket (dynamic pose): (**a**) front view; (**b**) lateral view; (**c**) back view.

Based on the analysis of Figure 13, the following conclusions were drawn:

- The horizontal free folds on the back of the back were removed;
- The lapels were correctly placed;
- The product was balanced on the avatar;
- The faces had a correct position.

## 4. Discussion

The images of the product on the avatar indicated that the designed model was well fitted to the body (in static and dynamic positions) and accurately followed the stylistic details of the sketch.

The integration of a certain amount of the dynamic effect in the design was carried out for the ergonomic shapes of the product.

The Ab = 4.5 cm and Aars = 3.6 cm values provided the fitted jacket model with a balanced placement on the avatar of the body.

Changing the distribution of the chest allowance corresponding to the three important areas of the product (back, front, and side) allowed a better placement of the product, eliminating the oblique free folds on the back and lowering the pressure level from dark red to brick red (i.e., from a critical area to a less critical area).

Changing the width of the lapel improved the stability of the garment shape.

The ergonomic design method of the flat shapes of the jacket patterns that partially integrated the values of the parameters describing the dynamic effects on the dimensions of the body led to a final product that meets the comfort and free movement requirements (normal dynamical situation). The study was carried out by using a layered structure typical of a business casual jacket that included a base material, an outer material, and a lining (the thickness of the base material was 1.62 mm).

Clothing exerts pressure on the body to an extent that is directly influenced by the type of apparel, the size of the connected sections, the layer structure, and the physical and mechanical properties. The number of layers in different product sections was variable. For example, the upper section of the jacket was made of the following layers: the base material, the interlinings, the padding, and the lining. The front section of the jacket had the following layers in its structure: the base material, the interlinings, the chest panel, and the lining. At level of the waist, the rear section of the jacket consisted of only a layer of base material and lining.

## 5. Conclusions

The 3D visualization of the model on the avatar enabled the identification of design solutions based on the ergonomic criteria of clothing models for bodies that had atypical conformations or for particular outfits.

By integrating dynamic data into the design process of fitted fashionable garments, the resulting garment product will better meet the customer's requirements and will significantly increase their confidence and satisfaction degree (wellbeing).

The ability to visualize the level of pressure (i.e., mechanical stress) that developed in the layered structure provided useful information for the optimization of material combinations (depending on the product category) as well as for the criteria for determining the materials for manufacturing while considering the requirements for wearing comfort.

The algorithm that has been developed in the current study can be implemented in the creation departments of garment companies that use a CAD system with specific functions for the geometric design of the shapes of product elements, providing designers with the chance to:

- Create the desired product in the shortest possible amount of time;
- Diversify the range of models that are launched on the market;
- Reduce the validation time of the new prototype;
- Experiment with complex design solutions.

The paper will enrich the literature on ergonomics in the field of fashionable clothing.

## 6. Limitations of the Research and Future Perspectives

The main limitation of this paper is that virtual systems do not yet accurately reflect the behaviour of textile materials while being worn and the interactions between the layers of multilayered garment products.

The methodology described must be adjusted when designing a jacket with a different style (different from the classic one) and cut line for a specific body shape (posture, proportion, and conformation) and made of base materials with different weft and warp shrinkage coefficients than of the ones mentioned in this study.

The presented study could provide a basis for future analyses and operational validations regarding the optimal areas for constructive allowances (chest, waist, and hip), for other layered structures that are typical for a men's jacket (thickness of materials), and for other types of silhouettes besides the high-waisted one (semi-tight silhouette, straight, and wide). Future research will deal with different types of clothing, e.g., trousers, shirts, and different categories of wearers, such as women and children.

**Author Contributions:** All authors (M.L.A., S.O., I.D., S.D.I., E.C.L. and I.I.) were involved in the design of experiments and discussion of the results. All authors have read and agreed to the published version of the manuscript.

**Funding:** This research received financial support for publication of this article from "Gheorghe Asachi" Technical University of Iasi Romania (TUIASI).

**Institutional Review Board Statement:** Not applicable.

**Informed Consent Statement:** Not applicable.

**Data Availability Statement:** The datasets used and/or analysed during the current study are available from the corresponding author upon reasonable request.

**Conflicts of Interest:** The authors declare no conflict of interest. The funders had no role in the design of the study; in the collection, analyses, or interpretation of data; in the writing of the manuscript, or in the decision to publish the results.

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
