# Peer review of "A New Approach to Dynamic Anthropometry for the Ergonomic Design of a Fashionable Personalised Garment"

_sustainability, doi:10.3390/su14137602_

Round 1
Reviewer 1 Report
The manuscript entitled, i.e., a new approach to dynamic anthropometry for the ergonomic design of a fashionable bespoke garment, has been reviewed.
The manuscript is written with low scientific standards and lacks novelty. Thus, some suggestions are strongly recommended before further consideration of the manuscript in the worthy journal, i.e., Sustainability.
The introduction is not written concretely; I advise improving it significantly.
Since 3rd section is Results, please add a section related to Discussion.
Correct Figure numbering. Figure 2 is missing.
What were the key questions to be resolved by this study? Add them in the introduction section.
Many researchers have explored CLO3D, such as HU Jiaqi and SONG Ying, Research on the dressing effect evaluation of Qipao based on CLO3D virtual fitting technology, Journal of Silk, 2021, 58(12), 73-79, https://doi.org/10.3969/j.issn.1001-7003.2021.12.012
ZHANG Weimeng, MA Fang, Analysis of cross structure of Han Chinese clothing based on CLO3D platform, Journal of Silk, 2021, 58(2), 131-136, https://doi.org/10.3969/j.issn.1001-7003.2021.02.019
What is the novelty of this work? Many industries use CLO 3D. What is the novelty of this work? Consider reading and citing Hu and Zhang's work to find out the novelty of this work.
The conclusion is too long. Please be specific to the main conclusions.
What are the limitations of this research? Please add a section related to this research's future perspective or outlook and limitations.
Reviewer 2 Report
Dear Author(s),
The manuscript Sustainability 1705804 and titled “A new approach to dynamic anthropometry for the ergonomic design of a fashionable bespoke garment” was reviewed. It is regular paper considering design of bespoke garments.
The research aim is to explain the design principle of apparel product in particular male jacket case.
Paper is well written and it was quite explained for reader stand points. Findings from the research were acceptably discussed considering the literature, and conclusion was consistent with generated data.
It was evaluated that it is regular paper and finding could interest the readers. The manuscript can be accepted after some revisions.
- Please provide some stress equations related to bending, wrinkling and drape during product fit as discussed in the manuscript.
- Please also provise some basic shear base relations during fitting the parts for men jacket aplications, if possible.
My best regards,
Reviewer 3 Report
The authors propose a method to customize garments based on the customer's body dynamic.
The paper propose an interesting idea, although not new. Furthermore, it has different drawbacks. It needs major revisions. In the following I put a list of comments to address. Where possibile I will indicate the section or the lines in bold font before my comment. Verbatim from authors paper is reported in italic font.
Title: the title contains and refer to "bespoke". The proposal instead places itself on the Made-to-measure level. Authors, should:
- change the title or explain in which way it is bespoke
- add literature background on the differences between bespoke, made-to-measure, ready-to-wear and the several shades of each level for people not expert in the fashion and tailoring areas
Abstract (lines 32-35): too vague concepts. The authors must update the abstract in order to be more technically sound.
Introduction: the vast majority of the introduction is not backed by proper literature or quantitative data supporting what authors state.
Authors states (lines 163-) "[...] At the enterprise level, data regarding the shape and dimensions of the human body are collected by either 3D scanning or by direct measurement of a static subject.[...]"
Please provide examples of this.
Line 246: The customization Armani and other brands provide, is not bespoke, rather Made-to-measure or just Made-to-Order. Indeed, this is something to improve in the paper as I also suggested before.
Line 251: please provide the link to all those software and a brief description of commercial tools like those.
Figure 1: I understand is just an instance, but it does not represent a proper jacket. Proportions, button positions, length, etc are not proper.
[...] The jacket model chosen for this case study (Figure 1) was suitable as a current business outfit. It had a high fit, a classic cut, and two-piece sleeves. The product closed with a single row of buttons, and the top had a collar and lapels. The product had a pocket with a loop (on the top of the left side) and two pockets with flaps below the waist.[...]
Authors should provide a detailed description of what is a business outfit. furthermore, an outfit is usually composed of more elements than just a jacket.
Please explain what is a pocket with loop. To my knowledge it is referred as breast pocket.
Figure 6. Dynamic poses: are those actually performed by the participants/customers or not? If not, how they reflect the actual customer?
Overall:
- It is not clear to me how customers' measurements are taken, nor how has been done with the 50 participants. Has the measurements taken compared with those that a tailor takes? Are they the same?
- What are the impact of such technology? (needs a paragraph)
- Have the authors planned an operational validation?
- How the solution cope with sustainability? (needs a paragraph)
- The paper needs a dedicated section about related works and a point-to-point comparison with what has been done in the literature and/or at the commercial level.
Round 2
Reviewer 1 Report
All requested suggestions have been made.
Author Response
Thank you for your valuable suggestions!
Reviewer 2 Report
I accept the revised paper at current status.
Regards,
Author Response

(The authors gave the same response as above.)

Reviewer 3 Report
I appreciate the effort of the authors revising the paper: e.g., the abstract is more technical, they have tried to answer to all my questions. The paper has improved a lot since the first round. Still, it has some drawbacks:
General: the authors refer to business casual. Please report some examples (figures) from the fashion industry in the paper given it is somehow a new term grown mostly in the last 10 years. Dating back taxonomies include ceremony, formal, informal and sportive categories only.
Abstract (33-35): the reader cannot know what the acronyms Ab and Aars refer to.
Introduction: it should report the structure of the paper and the major contribution. Both should be placed and highlighted in a dedicated paragraph.
Other questions:
1. Have you considered how the cloth/fabric reacts to the cut and the usual steaming process? Is this accounted in your system?
2. In this paper, it seems that the problems affecting a jacket's fit are only related to measurements. To the best of my knowledge the fit is also affected by: iron pressing (done at high-end fashion levels and bespoke), typology of sewings (especially in the armhole area), padding of the canvas on the chest and lapels of the jacket, angle of the canvas (on the bias or not), type of the shoulder and armhole (open armhole and closed armhole lead to two different kind of shoulder line), and back collar shape, construction and sewing because it is the only piece of the jacket that "touches" all the others being sewn on the lapels, on the back of the jacket and the shoulders. Besides, another important point is related to the button stance which should be customized/personalized as well based on the wearer. Authors should discuss deeply about these concepts and to improve the limitations section.
Round 3
Reviewer 3 Report
I am satisfied with the answers provided by the authors.
Can't wait to see further improvements of this project!